# Initial Investigations of the Cranial Size and Shape of Adult Eurasian Otters (*Lutra lutra*) in Great Britain

**DOI:** 10.3390/jimaging6100106

**Published:** 2020-10-08

**Authors:** Damian J. J. Farnell, Chern Khor, Wayne Nishio Ayre, Zoe Doyle, Elizabeth A. Chadwick

**Affiliations:** 1School of Dentistry, Cardiff University, Heath Park, Cardiff CF14 4XY, UK; khorch@cardiff.ac.uk (C.K.); AyreWN@cardiff.ac.uk (W.N.A.); ChadwickEA@cardiff.ac.uk (E.A.C.); 2School of Biosciences, Cardiff University, Heath Park, Cardiff CF10 3AX, UK; DoyleZC@cardiff.ac.uk

**Keywords:** cranial variation, otters (*Lutra lutra*), 3D surface scanning, multivariate statistical methods

## Abstract

Three-dimensional (3D) surface scans were carried out in order to determine the shapes of the upper sections of (skeletal) crania of adult Eurasian otters (*Lutra lutra*) from Great Britain. Landmark points were placed on these shapes using a graphical user interface (GUI) and distance measurements (i.e., the length, height, and width of the crania) were found by using the landmark points. Male otters had significantly larger skulls than females (*P* < 0.001). Differences in size also occurred by geographical area in Great Britain (*P* < 0.05). Multilevel Principal Components Analysis (mPCA) indicated that sex and geographical area explained 31.1% and 9.6% of shape variation in “unscaled” shape data and that they explained 17.2% and 9.7% of variation in “scaled” data. The first mode of variation at level 1 (sex) correctly reflected size changes between males and females for “unscaled” shape data. Modes at level 2 (geographical area) also showed possible changes in size and shape. Clustering by sex and geographical area was observed in standardized component scores. Such clustering in a cranial shape by geographical area might reflect genetic differences in otter populations in Great Britain, although other potentially confounding factors (e.g., population age-structure, diet, etc.) might also drive regional differences. This work provides a successful first test of the effectiveness of 3D surface scans and multivariate methods, such as mPCA, to study the cranial morphology of otters.

## 1. Introduction

Geometric morphometrics is the field of the study of biological shape [1,2,3,4,5]. Such shapes (e.g., of whole organisms or faces) are often defined by a collection of measurements at, or between, a predefined set of anatomically recognizable “landmark points”. Subsequently, “superimposition” methods, such as Procrustes transformation/analysis, are used to correct for centering, orientation, and scale in order to provide shape variables [1]. Multivariate data contain more than one “outcome” variable; here the *x*-, *y*-, and *z*-components of the Cartesian landmark points. Multivariate statistical methods, such as principal components analysis (PCA) [1], linear discriminant analysis [6], and multivariate analysis of variance [6,7], provide us with ways to analyze such (often highly correlated) data. Multivariate regression models may be used be used to study geometric morphometrics [8]. 

However, another multivariate method that has previously been used to study human shapes in particular is given by multilevel principal components analysis (mPCA) [9,10,11,12,13,14,15,16], which is a generalization of PCA that allows for us to account for groupings or clusters in our population of shapes. Indeed, mPCA allows us to isolate (to some extent at least) competing effects at different levels of the model. mPCA has also previously been employed in active shape models in order to segment image features in the human spine [9]. The authors note that mPCA “offers more flexibility and “allows deformations” (i.e., changes in shape) that “classical statistical models cannot generate” [9]. We previously applied mPCA to investigate (in humans): facial shape changes by ethnicity and sex [10,11]; the act of smiling [12,13]; facial shape changes in adolescents due to age [14,15]; and, maternal smoking and alcohol intake on the facial shape of children [16]. Here, we wish to extend these calculations to study otter cranial morphology.

The Eurasian otter (*Lutra lutra*) (hereafter: otter) is a carnivore of the family Mustelidae, and it is native across much of Eurasia [17]. Within this distribution, genetic sub-structuring is evident at both the broad scale (e.g., across Europe [18]) and at smaller scales (e.g., within the UK [19,20,21]). As of yet, there has been relatively little exploration of potential associations between genetic and phenotypic variation, although craniometric differences between countries have been observed [22]. Eurasian otters are sexually dimorphic (males are larger), and it has been suggested that differences in skull morphology may allow dietary separation between the sexes [22]. Differences in diet that are associated with age, body size, and sex have been reported [23]. Eurasian otters are primarily opportunistic piscivores; in the UK, regional and temporal variation in availability of prey species is reflected in diet [23]. Therefore, long term spatial variation in otter diet might drive evolutionary adaptations of otter cranial morphometry. Previous investigations of otter cranial morphometry [22,24,25,26] have focused on physical measurements (distances and angles) rather than by using three-dimensional (3D) scans. 

Here, we aim to first demonstrate that 3D surface scans can provide measurements of cranial distances that are in good agreement with direct physical measurements of the crania obtained using a caliper. Secondly, we also wish to “prove the principle” that multivariate statistical techniques (i.e., mPCA here) can be applied to landmark points obtained from the 3D surface scans by using a graphical user interface. Finally, we wish to explore whether these distance measurements show any differences by sex and geographical area, thereby demonstrating the potential usefulness of such 3D scans in analyzing such cranial shapes. 

## 2. Materials and Methods

### 2.1. Shape Acquisition

The Cardiff Otter Project was established in 1992 to investigate the health and biology of otters in the UK. Otters found dead (largely road traffic casualties) were collected, and then stored frozen at −20 °C prior to postmortem examination. For each otter, location of origin was recorded by the finder, and a range of biometric data (including sex, age-class, length, and weight) were recorded during a standardized postmortem examination (see www.cardiff.ac.uk/otter-project). Skeletal material, including the skull, was retained, and subsequently cleaned and archived by the National Museum of Scotland. For this study, 59 adult otter crania were selected in order to give a balanced sex ratio and broad geographic coverage (sex: 31 male, 28 female; geographical area: 21 Wales, 13 SE England; 15 SW England, 10 north England and Scotland; assigned according to genetic groupings defined by [21]). 

3D scans of the upper part of the otter crania were obtained using a (dental) Renishaw Medit T300 (blue light) scanner. The quoted accuracy of scans for the blue light scanner by the manufacturer (Renishaw, Wotton-under-Edge, UK) is 56 µm. Only partial scans were attainable due to the size of the crania; these partial scans were “stitched together” using MATLAB R2019b, as illustrated in Figure 1. 

A graphical user interface (GUI) (Meshmixer 3.5.474) was then used to place 31 3D landmark points for each otter cranial shape file, as illustrated by Figure 2. Cranial distance measurements, including the length, height, and width of the otter crania, were performed. These were found first by using these landmark points (referred to as “GUI-based” distance measurements) and separately by using direct “physical” measurements (referred to as “physical” distance measurements) on the crania by using a digital caliper (maximum precision of ~0.01 mm, in principle). 

Sets of landmark points (defined above) that were represented by the shape vector z were centered and aligned in 3D to the mean shape before analysis. This was carried out by using point cloud registration in MATLAB, which produces a rigid transformation of each shape with respect to the overall mean shape. This process corrected for centering and alignment, although not scale. Procrustes “superposition” was not carried out here.

Uncertainty in the positions of landmark points can occur, because of scanning inaccuracies, merging of point clouds, and finally in point placement. The stated accuracy of the Renishaw Medit blue light scanner is 56 µm, as noted above. The root mean square error (rmse) from point cloud registration in MATLAB, prior to merging of points clouds, was ~2 mm here. However, rmse is not a reliable estimate of the true of error that is involved in merging point clouds, as there is not a perfect one-to-one correspondence between these sets of points. An estimation of errors in manual placement can be achieved by repetition of the entire process of point placement for all shapes, although this was not carried out in this initial study. Instead, we examined the level of agreement between the two sets of landmark points (physical and GUI-based), while using intra-class correlation (ICC) coefficients in SPSS V25 (“single measures”) and mean “absolute” differences (MAD = |physical − GUI|; mean evaluated over all subjects). 

### 2.2. Statistical Analysis

Descriptive statistics were used for exploring the distance measurements initially as a function of sex and geographical area. Distributions of all cranial distance measurements were checked, and found to be normally distributed. Differences in size between the two sexes were analyzed via (unpaired) *t*-tests, and one-way ANOVA was used to test for size differences between geographical areas. Two-way ANOVA was used to simultaneously test for associations with sex and geographical area; their interaction could not be tested due to sample size limitations.

Thereafter, single-level PCA and multi-level PCA (mPCA) were applied to test for differences in shape. Analyses were repeated, on (i) data that were scaled in size to that of the mean, thus removing size variation (this dataset will be referred to as the *scaled* shape data), and on (ii) data that were not scaled (referred to as the *unscaled* shape data). Therefore, we were able to focus on associations with size and shape (*unscaled* data), and shape only (*scaled* data) in separate analyses. Linear discriminant analysis (SPSS V26) was also carried out on for the distance measurement data as yet another comparison to our results from mPCA.

PCA is the process of computing the principal components (also referred to as “modes of variation” here) that reflect the variation occurring in the data. The first principal component provides a direction that contains the highest amount of variance of the data, the second principal component provides another (orthogonal) direction that contains the second highest amount of variation of the data, and so on for all subsequent components. The magnitude of variation of each component is represented by its corresponding eigenvalue. PCA is therefore often used as a dimensional reduction technique. Here, PCA is carried out by forming a covariance matrix with respect to the landmark coordinate components, and this matrix is diagonalized in order to find eigenvalues and eigenvectors (i.e., the modes/components). New shapes may be readily fitted to a model provided by a weighted sum of components. These component weights are also referred to as “component scores” here and they may be standardized readily. Effectively there is only one level for “standard” PCA, and so we refer to this as single-level PCA. The mathematics of PCA are presented in the Appendix A. mPCA [7,8,9,10,11,12] allows us to decompose specific influences at specific levels of the model, i.e., sex and geographical area here, as illustrated by Figure 3. Covariance matrices are found at each level of the model and PCA is carried out for each of these matrices separately. Single-level PCA often mixes the effects of different influences in the principal components together. By contrast, mPCA is more likely to isolate specific influences at specific levels of the model, which is a strong advantage of mPCA over single-level PCA. Component scores may be found at each level of the model when fitting to new shapes and these scores may again be standardized readily. Again, the mathematics of mPCA are presented in the Appendix A.

Note that PCA also lies at the heart of active shape models (ASMs) and active appearance models (AAMs) [27,28,29,30,31,32,33,34], which are common techniques in image processing that are used to search for specific features or shapes in images, although ASMs and AAMs are not the focus of this article. Note that all of the calculations presented here for single-level PCA and mPCA were carried out using MATLAB R2019b, whereas statistical tests were carried out using SPSS V25.

## 3. Results

ICC coefficients between GUI-based and physical distance measurements were found to be high, i.e., ICC = 0.99, 0.85, 0.96 for length, width, and height (see Figure 2). The ICC coefficients indicated statistically significant (*P* < 0.001) levels of agreement between physical and GUI measurements. Mean absolute differences (again: MAD = |physical − GUI|) for the distance measurements were of order ~1 mm (minimum MAD = 0.66 mm and maximum MAD = 1.61 mm). These (small) differences between physical and GUI measurements were probably due to difficulties in consistently identifying points on the 3D surfaces (i.e., the “point correspondence” problem) for both sets of data (i.e., physical *and* GUI) rather than problems due the stitching process shown in Figure 1. Overall, these results show that good agreement occurred between physical and GUI-based measurements and that the point placement of landmark points was generally accurate. This agreement between the physical and GUI-based distance measurements is also demonstrated by descriptive statistics that are given in Table 1 for males and females separately. The results for these distance measurements in Table 1 also indicate that male otters have significantly larger (*P* < 0.001) crania than females in terms of length, height, and width.

Table 2 shows the results for the length, width, and height measurements of otter crania by geographical area in Great Britain. All of the distance measurements consistently indicate that the crania sampled from SW England were smaller than those from other areas (Table 2). Despite low sample sizes per group, significant differences (one-way ANOVA, *P* < 0.05) occurred in width and height, but not length. Two-way ANOVA indicated that significant (*P* < 0.05) differences occurred in length, width, and height with respect to sex and geographical area. Again, GUI-based distance measurements (not presented in Table 2) were found to agree well with physical measurements. 

Figure 4 shows the results for the eigenvalues from mPCA and single-level PCA. We see that results of mPCA are of the same magnitude and follow a similar pattern to those results of single-level PCA for both *scaled* and *unscaled* shape data (Figure 4). The largest eigenvalues for the *unscaled* data occurs at level 1 of the model (sex), whereas the largest eigenvalues for the *scaled* data occurs for level 3 of the model (between subjects). The results of mPCA on the *unscaled* data (exploring both size and shape differences) indicate that level 1 (sex), level 2 (geographical area), and level 3 (“between subjects”) contribute to 31.1%, 9.6%, and 59.3% of shape variation, respectively. The results of mPCA on the *scaled* data (exploring shape differences only) indicate that level 1 (sex), level 2 (geographical area) and level 3 (“between subjects”) contribute to 17.2%, 9.7%, and 73.1% of shape variation, respectively. 

The results for the first major mode of variation at level 1 (sex) via mPCA shown in Figure 5 for the *unscaled* data show strong changes in size (and not shape). These results are best illustrated by only considering those 17 points on the bottom of the otter crania, schematic also shown in Figure 5 as a reference. We remark that similar changes in size are seen for all points and also in the frontal (*yz*) and side (*xz*) planes. The results presented in Figure 5 support those results for the distance measurements that are shown in Table 1, which indicated that males have larger crania that females (e.g., in length of the skull of order ~7 mm). 

The results for the first major mode of variation at level 1 (sex) via mPCA shown in Figure 6 for the *scaled* data show some possible residual changes in size, but now also some subtle variations in shape. These results are again best illustrated by only considering those 17 points on the bottom of the otter crania (a schematic is also shown in Figure 6 as a reference). However, modes of variation are hard to interpret based purely on key landmark points. Again though, the results that are shown in Figure 5 and Figure 6 suggest broadly that changes in size and shape can occur as a function of sex.

The results for the first major mode of variation at level 2 (geographical area) via mPCA for the *unscaled* data show changes in size (and shape) also, which is in agreement with those results for the distance measurements shown in Table 2 that indicated that otters sampled from SW England had smaller skulls than those from the other areas. Furthermore, we believe that the first mode at level 3 might reflect changes in height to length (and width) ratio. However, any such changes at levels 2 and 3 are more subtle than those changes in shape that were observed at level 1 (sex). We note again that modes of variation are hard to interpret based purely on key landmark points and so the results at levels 2 or 3 are not presented here in this initial study. We believe that a clearer explanation of modes of variation would hopefully be aided in future studies by using larger sample sizes and “denser” point clouds (i.e., more landmark points) than are used here. The results of modes 1 and 2 via single-level PCA for the *unscaled* shape data (not shown here) are reminiscent of the first modes of variations at level 1 (sex) and level 3 (between subjects) via mPCA, as shown in Figure 7. However, it is probable that mixing of different effects (sex, geographical area, etc.) occurs in single-level PCA. The results for modes of variation for both mPCA and single-level PCA for the *scaled* data (also not shown here) demonstrate differences in cranial shape by sex and geographical area (etc.), although these modes are even harder to interpret than for the *unscaled* data. However, it was noticeable that large changes in size are not seen in any of the modes via either mPCA or single-level PCA for the *scaled* data. Larger sample sizes and “denser” point clouds (i.e., more landmark points) are again required to understand these subtle effects.

Figure 8 provides the results for standardized component scores for the *unscaled* data for mPCA. We see that strong clustering by sex is seen at level 1 (sex) in component 1 via mPCA, as expected, and that some differentiation between groups by geographical area is seen at level 2 in components 1 and 2 via mPCA. Indeed, component 1 at level 2 via mPCA separates SW England from the other areas, and component 2 differentiates north England from SE England. Intriguingly, there is strong overlap between Wales and SE England. However, we must be careful not to over-interpret these initial results, because the sample sizes are low in these initial investigations, especially for the analysis by regional area. No strong differences in standardized component scores by sex are seen at levels other than level 1 and, similarly, no strong differences are seen by geographical area at levels other than level 2, which is what we would expect if mPCA were correctly isolating specific influences at specific levels of the model. Therefore, this is an excellent check of our results. The results for standardized component scores via single-level PCA shown in Figure 8 show evidence of clustering by both sex and geographical area in both the first *and* second modes, which suggests that the effects of these factors might be mixed together. Furthermore, there is much more overlap in these component scores than that observed for scores via mPCA. Linear discriminant analysis applied to manual distance measurements also showed strong clustering by both sex and geographical area. Although a full treatment lies beyond the scope of this article due to small sample sizes, it is encouraging that another method (in addition to single-level PCA) provides support to the results of mPCA.

Similar patterns of strong clustering by sex at level 1 and geographical area at level 2 also occur for the *scaled* shape data, as shown in Figure 9. These results demonstrate that differences also occur between males and females and between geographical regions purely in terms of shape only. Some overlap again occurs between Wales and SE England. Again, no strong difference by sex is seen at levels other than level 1 and no strong differences are seen by geographical area at levels other than level 2. The results for components scores for single-level PCA results that are shown in Figure 9 show evidence of clustering by sex and geographical area, although, again, there is evidence of mixing of different effects in the first two modes and standardized scores have more overlap than seen in the scores via mPCA.

## 4. Discussion

The cranial shape of adult Eurasian otters (*Lutra lutra*) in Great Britain was investigated in this paper. Distance measurements found while using a GUI for the 3D scans had high ICC coefficients when compared to direct physical measurements on the crania that were found by using a caliper. This result demonstrates that good agreement occurred between the results of 3D scans and GUI-based landmark placement, which was the first aim of this study. They also present a successful initial test of the possible usefulness of such 3D scans in analyzing the cranial shape of otters. The use of imaging for cranial morphometrics (rather than physical measurement) eliminates the need to transport potentially fragile skulls for analysis, something that is particularly advantageous with geographically widespread species. The creation of a digital archive of data also provides significant legacy value for future analysis. Three-dimensional (3D) scanning of otter crania have other advantages over direct physical measurements, in principle, including reliability, time, and cost. However, a crucial first step is to show that the accuracy of such scans (and specifically landmark point extraction here) is at least as good as direct physical measurements, which we believe that we have demonstrated here. Probably the most important disadvantage of 3D scanning is that specialist (possibly expensive) equipment is needed in order to carry out the scanning, as well as to store and process the data. Direct physical measurements are also clearly a simpler and more straightforward, albeit more time-consuming, approach. Although structured light scanning, such as the blue light scanning used in this study, offers advantages over other 3D scanning methods (e.g., laser scanning), it suffers from potential imaging artefacts that arise from highly reflective or translucent materials. This, however, was not an issue given the opaque and matt surfaces of the otter crania.

We also wished to show that mPCA can be applied to study landmark points on the cranial shapes that were obtained while using a graphical user interface. mPCA indicated that sex and geographical area explained 31.1% and 9.6% of shape variation in *unscaled* shape data and that they explained 17.2% and 9.7% of variation in scaled data. Because there was an increase in the percentage variation from 17.2% for the *scaled* data (i.e., variations due to shape only) to 31.1% for the *unscaled* data (i.e., variations due to both shape and size) by sex, we interpret this as meaning that sex might influence both size and shape. By contrast, there is little change in the percentage variation for the *scaled* data (9.7%) as compared to the *unscaled* data (9.6%) by geographical area, which we interpret as meaning that geographical area might only affect shape. However, larger sample sizes and visualizations of dense point clouds are needed in order to confirm this statement, which lies beyond the scope of this article. The first mode of variation at level 1 (sex) of the mPCA model for the *unscaled* data also clearly corresponded to changes in size, as expected. This result in particular is very encouraging and it is an excellent validation of the mPCA method in these initial studies. Some changes in size were also seen in the first mode of variation at level 2 (geographical area) of the mPCA model for the *unscaled* data, as well as subtle shape variations. An advantage of mPCA over single-level PCA is that eigenvectors must be orthogonal within each level, but they do not necessarily have to be orthogonal between levels. Therefore, specific influences or factors should be more effectively isolated at specific levels of the model, as this should, in principle, reduce the effects of the common problem in PCA that leads to mixing of different effects in components if they are not orthogonal “in reality”. However, it has been remarked that between-groups PCA [35] (a form of two-level mPCA) can overestimate differences between groups when sample sizes are small, because between-group variation is represented well by differences between means, but within-group variation can be underestimated. Another limitation of mPCA is that the number of non-zero eigenvalues can be constrained by the number of groups at a given level. 

Clustering by sex and regional area was seen in standardized component scores via mPCA at appropriate levels of the models for both the *scaled* and *unscaled* shape data. Male otters were shown to have significantly larger skulls than females, as seen in other research [22,24,25,26]. Specifically, the quantitative results indicate that males had skulls that were 6.85 mm, 5.44 mm, 1.66 mm larger (*P* < 0.001) in terms of length, width, and height for adult males when compared to females. Strong differences in cranial size were also observed by geographical area in Great Britain that were often significant. We speculate that these results might reflect previously observed clustering by genetic profile in different regions of Great Britain [21]. We must also be careful when interpreting differences between groups to remember that errors in landmark point position could not be completely removed. These errors were due to the resolution of the 3D scans, merging of point clouds, and placement of the landmark points. However, the magnitude of differences between physical and GUI-based measurements, which we take here as a proxy for overall landmark error, had a minimum equal to 0.7 mm and maximum equal to 1.6 mm. These values are smaller than differences between sexes (e.g., of order 7 mm for the length) or between geographical areas (e.g., of order 5 mm for the length). Indeed, the natural variation between otters for the length, width, and height within each sex and geographical area group were of order (standard deviations) 1 mm to 3 mm. Furthermore, we note that there are many other factors (e.g., age, feeding habits) that might also affect cranial shapes. Our initial sample sizes were too small to robustly explore these additional variables. Despite this, the sample sizes probably were sufficiently large (i.e. around 30 per group) for comparisons between males and females; apparent spatial differences merit further investigation.

## 5. Conclusions

This study provides a successful first test of the effectiveness of 3D surface scans and multivariate methods, such as mPCA, to study cranial morphology, as well as suggesting some intriguing differences in cranial morphology among the UK otter population. Future studies will concentrate on larger sample sizes and developing and applying multivariate methods that can account for continuous covariates, such as age, as well as discrete ones, such as sex (etc.), for example, using (multilevel) partial least-squares methods.

## Figures and Tables

**Figure 1 jimaging-06-00106-f001:**
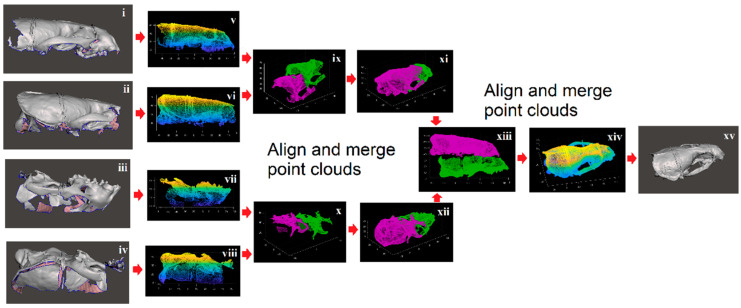
Stitching process of partial scans: point clouds extracted from shape files from partial three-dimensional (3D) scans of the crania on the left-hand side of the figure are aligned and merged (as shown) in order to form a complete surface shape file shown on the right-hand side of the figure. Four partial scans of the crania as shown (**i** to **iv**) were found for each otter. The original shape files (STL format) were used to generate point clouds (**v** to **viii**). The front and rear of the top and bottom sections were aligned (**ix**,**x**) and combined (**xi**,**xii**) by using the “point cloud register” command in MATLAB R2019b, and the resulting top and bottom sections were aligned and combined (**xiii**) to create a complete representation of the surface shape as a point cloud (**xiv**). MESHLAB V2016.12 (www.meshlab.net) was then used to create the final shape file (in STL format) from this point cloud (**xv**).

**Figure 2 jimaging-06-00106-f002:**
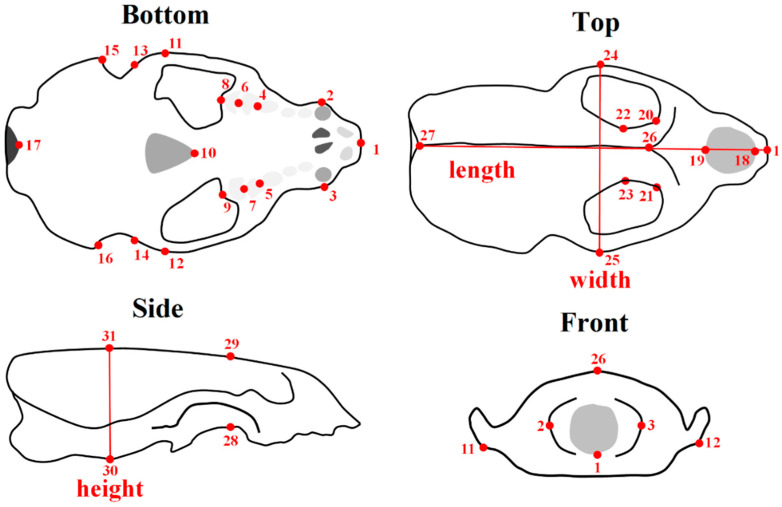
Schematic of the upper part of an otter cranium from different viewpoints with 31 landmark points indicated. Distances, such as the length, width, and height, of the skull can be found using these landmark points placed by using a graphical user interface (GUI) (referred to as “GUI-based” distance measurements). These distances were also measured directly for the crania by using a digital caliper (referred to as “physical” distance measurements.).

**Figure 3 jimaging-06-00106-f003:**
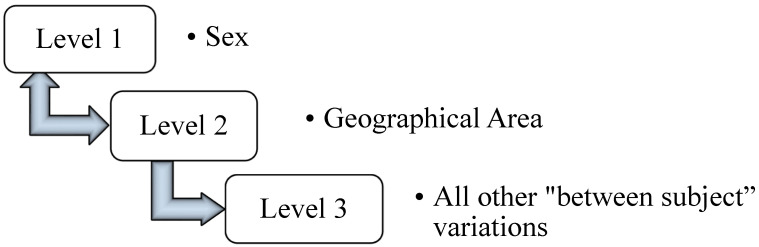
Schematic of the multilevel model used here. This is a non-nested model, i.e., there is no natural “nested order” to sex and geographical area (shown by the “double arrows” appropriately).

**Figure 4 jimaging-06-00106-f004:**
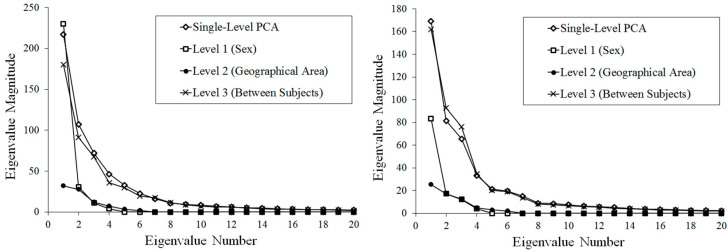
Eigenvalues from single-level principal components analysis (PCA) and multilevel Principal Components Analysis (mPCA): (**left**) *Unscaled* shape data; (**right**) *scaled* data (i.e., all shapes were resized to a common length scale).

**Figure 5 jimaging-06-00106-f005:**
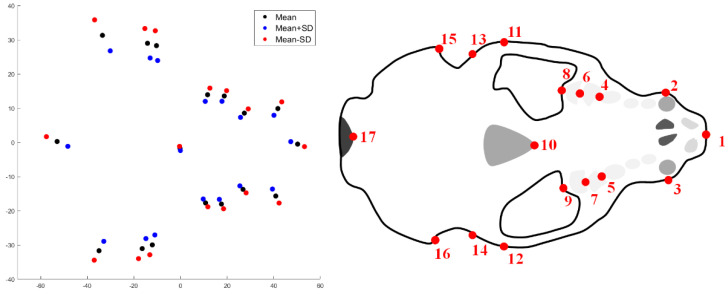
Sexual dimorphism in cranial size illustrated using mPCA (left-hand figure) for *unscaled* shape data for 17 points on the bottom of the crania (schematic shown again in the right-hand figure for the sake of comparison only). Blue dots represent females (mean + SD) red dots represent males (mean − SD). (Note that axes are measured in mm.).

**Figure 6 jimaging-06-00106-f006:**
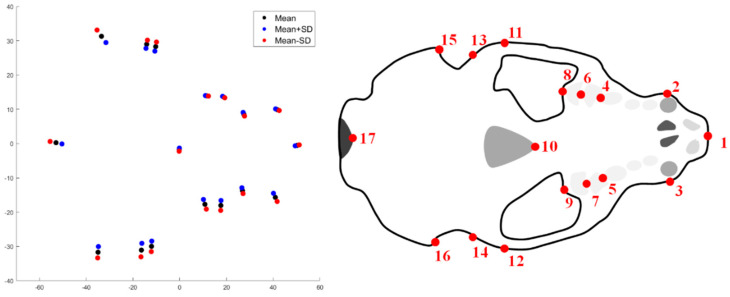
Sexual dimorphism in cranial size illustrated using mPCA (left-hand figure) for *scaled* shape data for 17 points on the bottom of the crania (schematic shown again in the right-hand figure for the sake of comparison only), which is more subtle in this case. Blue dots represent females (mean + SD) red dots represent males (mean − SD). (Note that axes are measured in mm.).

**Figure 7 jimaging-06-00106-f007:**
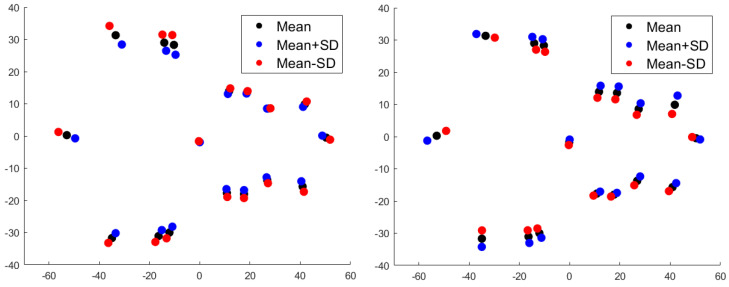
Results for the first two modes of variation using single-level PCA for *unscaled* shape data for 17 points on the bottom of the crania (**left**: first mode; **right**: second mode). Both modes show some evidence of changes in size, although other subtle changes in shape might occur also, especially for the second mode (Note that axes are measured in mm).

**Figure 8 jimaging-06-00106-f008:**
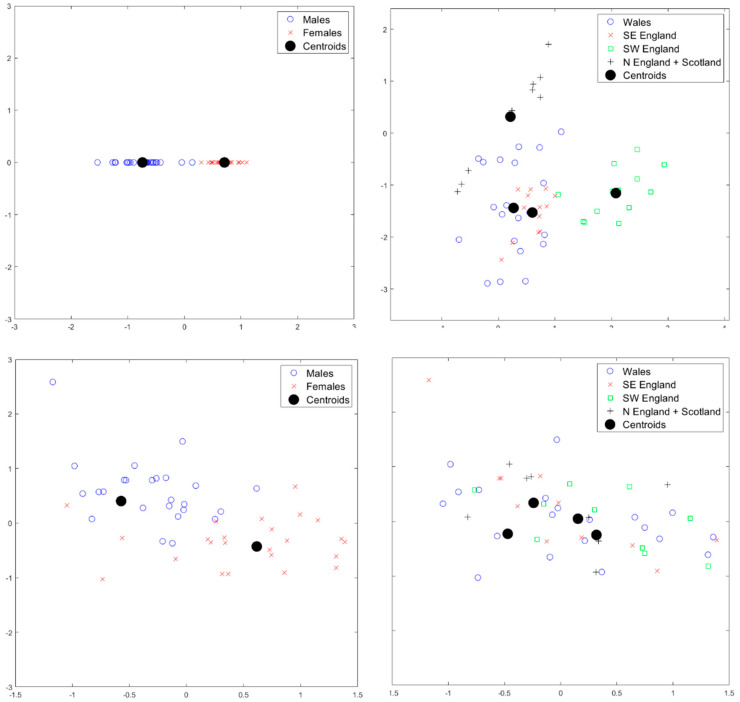
Results of mPCA for standardized component scores (*x*-axis: component 1; *y*-axis: component 2) for the *unscaled* shape data. Results of mPCA with $m_{1}=1$; $m_{2}=3$; $m_{3}=20$ in Equation (A4) are shown in the top row: (**left**) level 1, which shows a strong clustering by sex; (**right**) level 2, which shows clustering by geographical area. Results of single-level PCA are shown in the bottom row: (**left**) symbols chosen by sex; (**right**) symbols chosen by geographical area.

**Figure 9 jimaging-06-00106-f009:**
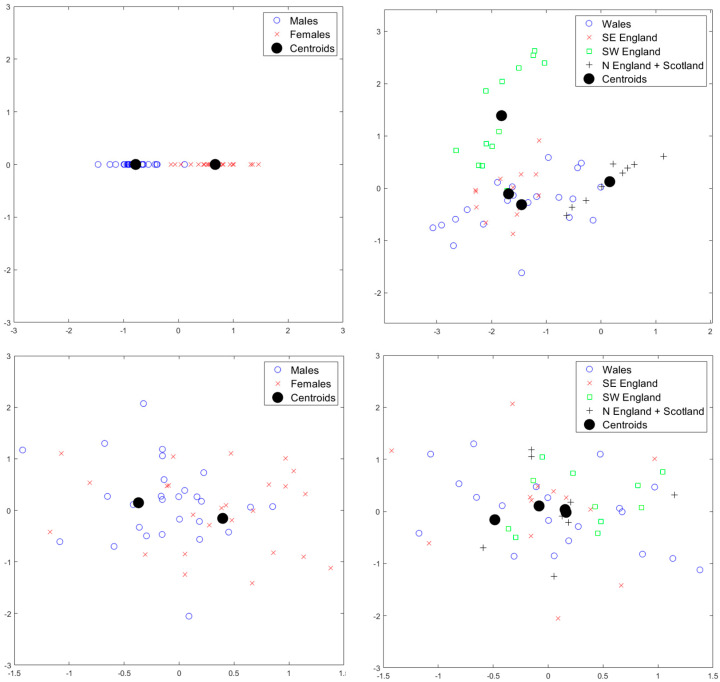
The results of mPCA for standardized component scores (*x*-axis: component 1; *y*-axis: component 2) for the *scaled* shape data. Results of mPCA with $m_{1}=1$; $m_{2}=3$; $m_{3}=20$ in Equation (A4) are shown in the top row: (**left**) level 1, which shows a strong clustering by sex; (**right**) level 2, which shows clustering by geographical area. Results of single-level PCA are shown in the bottom row: (**left**) symbols chosen by sex; (**right**) symbols chosen by geographical area.

**Table 1 jimaging-06-00106-t001:** Length, width, and height distance measurements of otter crania. Male crania are significantly larger than female crania via unpaired *t*-tests (*P* < 0.001). Excellent agreement is seen between direct, physical and GUI-based results for these distances.

Sex (Measurement Type)		Length	Width	Height
**Male (Physical)**	Mean (mm)	100.53	70.53	41.30
	SD (mm)	3.96	2.89	1.62
**Female (Physical)**	Mean (mm)	93.68	65.09	39.64
	SD (mm)	3.26	2.14	1.36
**Male (GUI)**	Mean (mm)	100.38	70.10	41.84
	SD (mm)	3.72	3.57	1.36
**Female (GUI)**	Mean (mm)	93.44	63.92	39.14
	SD (mm)	3.25	2.42	1.22

**Table 2 jimaging-06-00106-t002:** Length, width, and height distance measurements of otter crania (physical distance measurements shown here only). Crania from SW England are smaller than those from other areas. (Results from one-way ANOVA are also quoted in this table.).

Region		Length	Width	Height
**Wales**	Mean (mm)	97.23	67.00	40.76
	SD (mm)	4.08	3.10	1.45
**SE England**	Mean (mm)	99.03	69.21	41.14
	SD (mm)	6.57	4.98	1.95
**SW England**	Mean (mm)	94.63	66.57	39.42
	SD (mm)	4.34	2.90	1.45
**North England/Scotland**	Mean (mm)	99.08	70.75	40.81
	SD (mm)	4.17	2.39	1.75
ANOVA		*F* = 2.563; *df* = 3, 55; *P* = 0.064	*F* = 3.991; *df* = 3, 55; *P* = 0.012	*F* = 5.386; *df* = 3, 55; *P* = 0.03

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
