# Peer review of "Initial Investigations of the Cranial Size and Shape of Adult Eurasian Otters (Lutra lutra) in Great Britain"

_2313-433X, 2020, doi:10.3390/jimaging6100106_

Round 1

Reviewer 1 Report

The submitted paper is of very high quality in the sense of clarity of explanation, it is easy to understand, and it is generally on a high formal level. The introduction section provides sufficient background, and the research is designed appropriately as well.

The part of a paper describing the analysis of given data is without any single point to be improved, the use of statistical methods is well designed, well described and the results are clearly explained and discussed as well.

My only comment is on the part of a data acquisition, which, in my opinion, needs to be further extended.

The claim that the 3D scanning method is reliable and accurate (lines 56 and 313) is not sufficiently scientifically substantiated on the basis of the results obtained. From the results in a paper, we can assess it only as a similar to calliper measurement. We also cannot properly address the source of deviations between both methods (line 199).

I propose that authors should evaluate the overall uncertainty of the resulting coordinates of landmarks which are further mathematically processed, especially, the uncertainty of 3D scanning itself and the uncertainty of merging partial point clouds. The uncertainty of identifying the landmark points may be omitted since it has a similar influence in both methods to compare. This analysis will prove the accuracy of a method and point out its benefits. It should be taken into account when interpreting the results as well.

Since one of the aims of this paper is to introduce 3D scanning as a new acquisition method, I miss a clear comparison between the novel method (3D scanning) and the standard method (calliper) too. Authors declare high ICC indicating the methods might be of similar precision and briefly comments some advantages (line 318). However, there are other benefits and also disadvantages of 3D scanning, which needs to be further discussed.

Generally, the paper is very good, and I recommend to accept it after minor revisions mentioned above.

Author Response

Please see attached document for details of our response.

Reviewer 2 Report

As written in the title, this study is an initial analysis of the cranial shape variations among the species Lutra lutra, in order to foresee if the visualized variations in shape are due to sexual dimorphism or geographical clustering.

The paper is well written, although the methodological part (i.e., statistical analyses) is heavy and should be accompanied by the script or the code (which software did you use? Matlab? R could be a good option).

I have no major corrections to demand, but I suggest the authors to provide the methodological part some clarifications.

Statistical analyses

I definitively had a hard time with this part of the manuscript. As a matter of fact, the description of the steps of the data preparation and treatment was sometimes difficult to read and many commonly used procedures, such as Procrustes superimposition, are not indicated in the text, even if they were used if I understood well. Moreover, the mPCA is not common and I am not sure I understood how it worked. It could have been useful to have a first glance at the PCA results, to visualise the shape diversity within the morphospace. Then, most common and perhaps simpler methods to discriminate sex or geography could have been used, such as discriminant analysis on allometry-corrected shapes.

Discussion

ligne 328, you write that "geographical area might affect shape only". Why is that?

Author Response

(The authors gave the same response as above.)

Reviewer 3 Report

The manuscript of Farnell and coauthors reports some initial results on the sources of morphological variation in Otters. The paper has some merits such as the use of multi-level PCA but fails to properly set up the aim of the study in the scope of Journal of Imaging. The introduction is particularly representative of that missed target with a start on the ecology of otters, then without any evident link on the use of geometric morphometrics to study integration and modularity (which is not the aim of the paper at all), then a sort of justification of the submission to an image processing journal... In my opinion, the introduction should be  rewritten totally. Clearly the usefulness and accuracy of 3D scans in anatomy and in virtual collection is not new, people have even published about how to merge heterogeneous data from such collections.

Analysis of shapes has a long history of mathematical foundation. I don't understand if they use it or not. They use some kind of superimposition but it is clearly difficult to say if it's a Procrustes superimposition or not. Papers have been also published on how to merge size and shape (see for ex. review of CP Klingenberg on allometry in Dev. Genes and Evolution). All this background is not used or re-developed but it is difficult to tell. Instead of explaining all of that, the authors spend some lines (121-123) on digression on active shape models which I don't understand and they say themselves that it is not really relevant "not the focus of this article". 

Differences were tested using t-test (doesn't need to specify unpaired here) and one-way ANOVA but a proper modeling should be a two-way ANOVA setting or not the interaction to 0 given the sample size.

In the discussion, it is stated that geography might affect shape only based on the equivalence of the percentage of variance. Size could be analyzed on its own so this could be answered instead of speculated.

Minor comment:

  • L127 points. Do you mean individuals/observations. Points is quite misleading with landmarks
  • using z for the predicted phenotype in eq.2 and as the observed one in eq1 and eq.3 is misleading since the residual is not included in eq2.
  • L172-L173. Repeat from line 148.
  • L176. with respect to an appropriate cost function. Which one ?
  • Table 2. The ANOVA table (F + dll + pvalue) instead of just the pvalue is better.
  • Figure 7 and Figure 8. The sex panel. As the sex is a single dimensional effect it could not be represented as a bivariate plot as the second dimension should have a null variance. Choose density plot or histogram. Actually it is something I don't understand in your analysis that eigenvalues for the sex level are higher than 0 for axes after the first (Figure 7).

Author Response

(The authors gave the same response as above.)

Round 2

Reviewer 3 Report

The authors answer many of my initial concerns by rewritting their introduction and explaining with more details their approach.